# Fatigue Performance of an Additively Manufactured Zr-Based Bulk Metallic Glass and the Effect of Post-Processing

**Navid Sohrabi** [1,*], **Milad Hamidi-Nasab** [1,2], **Baptiste Rouxel** [1], **Jamasp Jhabvala** [1], **Annapaola Parrilli** [3], **Maurizio Vedani** [2] **and Roland E. Logé** [1]

[1] Thermomechanical Metallurgy Laboratory (LMTM), PX Group Chair,
Ecole Polytechnique Fédérale de Lausanne (EPFL), 2002 Neuchâtel, Switzerland;
milad.hamidinasab@epfl.ch (M.H.-N.); baptiste.rouxel@epfl.ch (B.R.); jamasp.jhabvala@epfl.ch (J.J.);
roland.loge@epfl.ch (R.E.L.)

[2] Department of Mechanical Engineering, Politecnico di Milano, 20156 Milan, Italy; maurizio.vedani@polimi.it

[3] Center for X-ray Analytics, Swiss Federal Laboratories for Materials Science and Technology (Empa),
Überlandstrasse 129, 8600 Dübendorf, Switzerland; annapaola.parrilli@empa.ch

\* Correspondence: navid.sohrabi@epfl.ch

**Abstract:** Fatigue is the most common cause of failure of mechanical parts in engineering applications. In the current work, we investigate the fatigue life of a bulk metallic (BMG) glass fabricated via additive manufacturing. Specimens fabricated via laser powder-bed fusion (LPBF) are shown to have a fatigue ratio of 0.20 (fatigue limit of 175 MPa) in a three-point bending fatigue test. Three strategies for improving the fatigue behavior were tested, namely (1) relaxation heat treatment, giving a slight fatigue life improvement at high loading conditions ($\geq$250 MPa), (2) laser shock peening, and (3) changing the build orientation, the latter two of which yielded no significant effects. It was found that the presence of lack of fusion (LoF) had the preponderant effect on fatigue resistance of the specimens manufactured. LoF was observed to be a source of stress localization and initiation of cracks. The fatigue life in BMGs fabricated by LPBF is thus primarily influenced by powder quality and process-induced defects, which cannot be removed by the post-treatments carried out in this study. It is believed that a slight increase in laser power, either in the near-surface regions or in the core of the specimens, could improve the fatigue behavior despite the associated (detrimental) increase of crystallized fraction.

**Keywords:** bulk metallic glass; fatigue; laser powder-bed fusion; laser shock peening; defects





## 1. Introduction

Recent progress in laser powder-bed fusion (LPBF), the most widely used additive manufacturing (AM) technique for metals, has led to a strong interest in the manufacturing of bulk metallic glass (BMGs) [1–15], since high local cooling rates induced by the LPBF process match the requirements for manufacturing BMGs with substantial dimensions and complex shapes [8]. This challenge generated a lot of enthusiasm due to the exceptional mechanical properties of BMGs, which make them possible candidates in a number of industrial fields, such as micro-mechanics, medicine, aerospace, and automotives. In addition to their excellent wear and corrosion resistance, the amorphous atomic structure of BMGs provides very high strength and elastic limits with a low Young modulus [16]. To the best of the authors' knowledge, no studies have been published yet on the fatigue life of BMGs produced via AM.

BMGs are surprisingly very susceptible to the damage caused by cyclic loading with respect to their very high strength [17]. The typical fatigue ratio (fatigue limit/ultimate tensile strength (UTS)) of BMGs is lower than that of crystalline low carbon steel or aluminum [18]. The fatigue limit of BMGs range from 150 MPa to 1050 MPa, which corresponds to fatigue ratios between 8% and 50% [19,20]. These properties have been shown to depend on

multiple factors such as the loading mode [20–22], the sample geometry [7,23], the testing environment [24,25], and the thermal history [26–28].

Fatigue failure mechanisms in metals usually comprise three stages: (I) the nucleation of a crack, (II) its propagation, and (III) the rapid fracture after the crack size has reached a critical threshold. Unlike for crystalline materials, in BMGs, cracks cannot nucleate at grain boundaries or from crystal irregularities [7]. Instead, they initiate from shear bands forming near defects and locally accumulating free volume [26,29,30]. When a fatigue crack is initiated, it grows and propagates, generating a typical fatigue striation pattern on the fracture surface. The striations are usually associated with alternative blunting and re-sharpening of the crack tip [24,31,32]. In BMGs, this blunting comes from a local softening generated by multiple shear bands at the crack tip and the associated free volume (observed by Doppler-broadening spectroscopy (DBS) [26]), and nano-voids [33,34]. Concerning the re-sharpening of the crack tip, Scudino et al. [32] stated "the characteristic compressive stresses required to re-sharpen the crack tip are developed in a BMG upon unloading". The coalescence of free volume into nano-voids is also potentially enhanced by the co-nucleation of nanocrystals [35–37]. Launey et al. [26] found the initial free volume actually had little effect, and rather the crack tip deformation redefined the local free volume and controlled the crack propagation.

One of the main drawbacks of the LPBF process (for crystalline and amorphous alloys) is the presence of defects, such as lack of fusion (LoF) porosity in the manufactured parts [38]. These defects impair the part integrity and reduce the mechanical properties such as tensile strength of a BMG [39,40], impact toughness [40], and fatigue life of crystalline alloys [41–45]. Yadollahi et al. [43] also investigated the effect of the building orientation on the fatigue life of the manufactured Ti-6Al-4V specimens. Due to the orientation of the LoFs with respect to the applied load, horizontal specimens can exhibit improved fatigue life compared to vertical specimens [46].

Another drawback of LPBF is the presence of a high level of tensile residual stresses (TRS) in the fabricated parts due to cyclic and localized rapid heating and cooling. TRS can be detrimental to the geometrical accuracy [47] and fatigue life [48] of the parts. In addition, TRS can induce cracking in metallic alloys (for instance, CM247LC [49]). Laser shock peening (LSP) [48,50] and shot peening (SP) [51] are two surface treatment processes that can generate compressive residual stresses (CRS) in the surface and sub-surface regions. CRS in the loading direction and strain hardening are known to be favorable for fatigue properties of crystalline alloys [48,50,51]. LSP and SP have also been tested on BMGs to improve the ductility of the samples by inducing additional free volume, although not homogeneously [52–54]. Zhang et al. [52] and Wang et al. [55] reported softening close to the surface after applying SP and LSP, respectively, because of the induced free volumes. Raghavan et al. [56] investigated the effect of shot peening on the fatigue life of a Zr-based metallic glass, and no significant improvement was reported. Gao et al. [57] reported that LSP had a more pronounced effect on fatigue life improvement for an Al-alloy than SP because of inducing deeper CRS and better surface finish. To the best of the authors' knowledge, no studies have been conducted on the effect of LSP on the fatigue life of BMGs.

Due to a high TRS level in the LPBF parts, Aboulkhair et al. [58] performed stress relief heat treatment on AlSi10Mg and reduced fatigue crack growth. Launey et al. [26] performed different heat treatments on an as-cast Zr-based BMG below the glass transition temperature, $T_g$, to relieve the residual stresses and reduce the free volume in the material. They showed that structural relaxation resulted in improved fatigue strength, attributed to the higher resistance in crack initiation with lower free volume, in agreement with earlier work by Launey et al. [27], and with the associated increase in hardness [28].

Oxygen is one of the detrimental impurities for BMGs. Keryvin et al. [59] studied the effect of oxygen content on the fracture toughness of a Zr-based BMG and showed that oxygen impurities deteriorated the fracture toughness of the sample by forming brittle dendrites. Best et al. [15,39] attributed the low fracture toughness of a Zr-based BMG

fabricated via LPBF to the high dissolved oxygen concentration in the samples. Wegner et al. [13] tried to reduce the oxygen content in AMZ4 samples fabricated via LPBF by using a reducing atmosphere ($Ar_{98}H_2$) during printing. However, the amorphous fraction of the sample fabricated in pure argon atmosphere was higher than in the reducing atmosphere at similar energy inputs, and no significant change in the flexural strength of the samples fabricated in the two conditions was observed. Therefore, the strategy of in situ decrease of the oxygen content during the LPBF process was not successful.

One of the main challenges in AM of BMGs is unwanted crystallization in the heat-affected zones (HAZs) due to cyclic heating and cooling [8,60]. Unwanted crystallization causes brittleness, with cracking typically associated with the combination of intermetallics and TRS [60]. Wang et al. [18] studied the effect of partial crystallization of three as-cast Zr-based BMGs on fatigue properties. Partial crystallization reduced dramatically the fatigue limit of BMGs because crystals were brittle and considered as "weak spots". Another study confirmed that partial crystallization of a Zr-based BMG significantly increased the fatigue-crack growth rate and drastically deteriorated the fracture toughness of the BMG [61].

All studies in the literature that investigated the density of AMZ4 samples fabricated via LPBF reported the presence of defects, such as spherical porosities and LoFs, in the final consolidated parts [8,9,13,39,40,62–64]. To the authors' knowledge, such defects can never be eliminated in commercial grade AMZ4 without leading to excessive crystallization. Despite the presence of defects, good static mechanical properties such as tensile, compressive, and flexural strength were achieved. However, they are still well below the values reported for the fully dense as-cast alloy [65].

In this study, we investigate the fatigue life of an industrial-grade Zr-based BMG (AMZ4) fabricated via LPBF and containing a substantial amount of oxygen. A low fatigue limit was achieved, which could be mainly attributed to the presence of high oxygen content and defects inherent to the LPBF process. Several strategies, such as LSP treatment, relaxation heat treatment (sub-$T_g$), and change of the build orientation, are tested to evaluate their corresponding effect on the fatigue life of AMZ4. It is shown that post-processing is not a solution bringing significant improvement. Although AMZ4 behaves reasonably well under static conditions, it is not suited for applications under cyclic loading, at least with the present (optimized) processing conditions.

## 2. Materials and Methods

In this study, the Zr-based metallic glass powder, under the trade name of AMZ4, was supplied by Heraeus GmbH (Hanau, Germany), with a nominal chemical composition of $Zr_{59.3}Cu_{28.8}Al_{10.4}Nb_{1.5}$ (at.%). LPBF processing was carried out using a TruPrint 1000 machine (TRUMPF) with a laser spot diameter of 30 µm in an inert gas (argon) atmosphere. The processing parameters were the same as those in our previous study [8]; laser power, scanning speed, hatching distance, and layer thickness were 30 W, 600 mm/s, 90 µm, and 20 µm, respectively. An island scanning strategy with an island size of $4 \times 4$ mm$^2$ with an orientation change of 90° after each layer was utilized for the manufacturing of the fatigue specimens.

Three-point bending specimens with a size of $60 \times 10 \times 3.5$ mm$^3$ were fabricated on a support structure with a height of 1 mm. Two different build orientations, horizontal and semi-vertical, were chosen, as shown in Figure 1. All specimens were fabricated with a horizontal orientation, except six of them in the semi-vertical orientation.

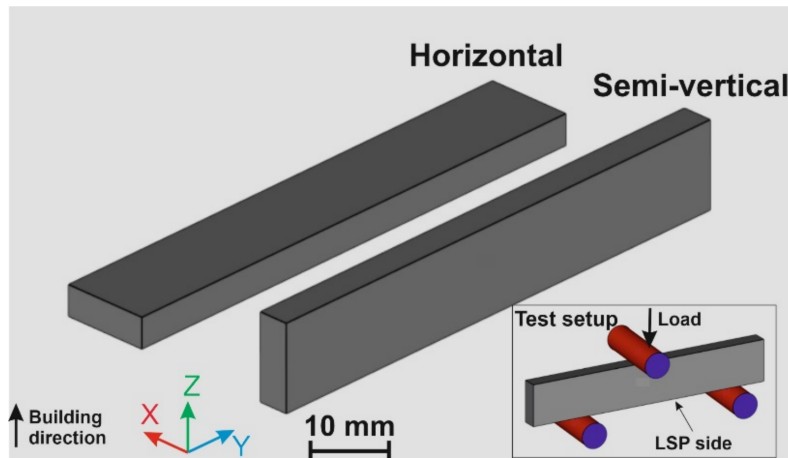

**Figure 1.** Fatigue specimens ($60 \times 10 \times 3.5$ mm$^3$) with two different building orientations, horizontal and semi-vertical. The inset shows a schematic of the three-point bending fatigue test.

Two faces ($60 \times 10$ mm$^2$) of the specimens were ground and mirror-polished until a thickness of 3 mm was reached. The faces in contact with the rolls ($60 \times 3$ mm$^2$) (depicted in the inset of Figure 1) of the three-point bending fatigue platform were ground with sandpaper up to a mesh size of 2500.

A field emission scanning electron microscope (FESEM; ZEISS GeminiSEM450) was used to investigate the microstructure and the fractography of the specimens after rupture. X-ray diffraction (XRD) tests were performed on the powder and fabricated parts to check the amorphous nature via a PanAlytical Empyrean diffractometer with Cu-K$\alpha$ radiation. Differential scanning calorimetry (DSC) tests were carried out in a DSC 8000 from Perkin Elmer using a heating rate of 20 °C/min in Ar atmosphere. To reveal the microstructure, the printed specimens were etched with a solution of 45 mL water + 45 mL HNO$_3$ + 10 mL HF at room temperature. Vickers' hardness tests (HV1) were carried out using a Qness Q10A machine with a 1 kg force and a dwell time of 12 s.

The relaxation heat treatment was performed on the printed specimens (RHT), prior to their removal from the substrate, at 320 °C (0.8 $T_g$) and for 1 h, in a Borel FP 1400 furnace in air. Subsequently, they were air-cooled to ambient temperature, out of the furnace. The oxidized surfaces were ground and polished.

A laser source of Nd:YAG SAGA HP-class (Q-switch) laser from Thales Laser company was used for LSP tests. LSP experiments were performed with water as the confining medium and without coating, using the parameters given in Table 1. The target surfaces were those under tension in the three-point bending fatigue test (shown in the inset of Figure 1).

**Table 1.** LSP parameters used in this study.

| Parameter | Value |
| --- | --- |
| Pulse Duration (ns) | 6.3 |
| Wavelength (nm) | 1064 |
| Frequency (Hz) | 5 |
| Spot Size (mm) | 1.2 |
| Energy Per Pulse (J) | 1.5 |
| Overlap of Spots (%) | 50 |
| Beam Spatial Energy Distribution | Top-hat |

The set of three-point bending samples is summarized in Table 2, with four different processing histories.

**Table 2.** Specimens for three-point fatigue tests.

| Processing History | Label | Number of Specimens |
| --- | --- | --- |
| As-Built | AB | 14 |
| AM + Relaxation Heat Treatment | RHT | 12 |
| AM + Laser Shock Peening_Horizontal Orientation | LSP-H | 13 |
| AM + Laser Shock Peening_Semi-Vertical Orientation | LSP-SV | 6 |

Three-point bending fatigue experiments were performed in load-controlled mode utilizing an MTS Acuman 3 electrodynamic test platform equipped with 3 kN of loading capacity. The experiments were performed on four sets of specimens with a load ratio, R, of 0.1 in stress-controlled mode with stress ranges varying from 160 to 350 MPa at an ambient temperature environment, such as to investigate fatigue lives between $4 \times 10^4$ cycles and the runout limit, which was set to $5 \times 10^6$ cycles. The test frequency was set to 30 Hz. The loading span length and the roller diameter were 40 mm and 5 mm, respectively.

Micro computed tomography (micro-CT) analysis was carried out using an EasyTom XL Ultra 230-160 micro/nano-CT scanner (RX Solutions, Chavanod, France). The scanner operated at 200 kV with a current of 65 μA. An Al–Cu filter was interposed between the X-ray source and the samples. The samples were scanned full 360° with a rotation step of 0.25° and frame average of 10. The nominal resolution was set at 2.5 μm voxel size. We reconstructed the scan images using a moderate beam hardening correction. The obtained tomographic cross-sections (16-bit TIFF format) were segmented for pores with the application of a global threshold to quantify: the porosity as a percentage of the bulk volume, the overall pore size distribution and LoF size distribution as pore equivalent diameter distribution, and the number of existing LoFs. To determine and select LoFs, the shape parameter was calculated in Avizo software using the following formula:

$$Shape = \frac{S^3}{36\pi V^2} \tag{1}$$

where *S* is the 3D surface area of the pores, and *V* is the volume of the pores. The shape of 1 is considered a perfect sphere, and an increasing shape number leads to more complex and convoluted shapes [66]. Therefore, a shape $\geq 5$ was considered as the threshold for detecting LoFs.

### 3. Results

The XRD pattern of the gas-atomized AMZ4 powder is presented in Figure 2a. No sharp peaks were detected, which is an indication of amorphous structure in the powder (within the detection limit of the XRD technique). Figure 2b,c show an SEM image and a DSC curve of the AMZ4 powder, respectively. Most of the powder particles were observed to be spherical. The DSC results are presented in Table 3. The combination of amorphous structure and spherical shape of the powder makes this alloy a good candidate for powder-bed additive manufacturing techniques.

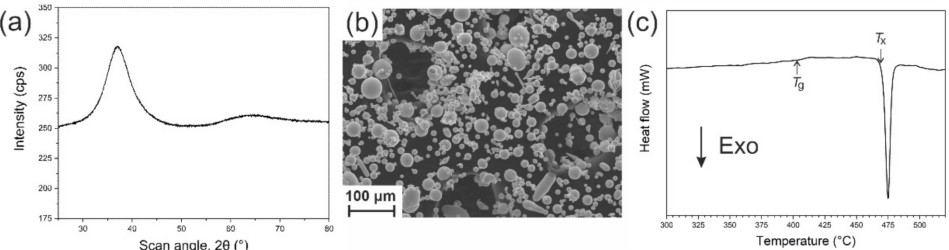

**Figure 2.** (**a**) XRD pattern of AMZ4 powder, which shows an amorphous structure (within the detection limit of the XRD technique), (**b**) SEM image of the powder particles, and (**c**) DSC curve of the powder measured with a heating rate of 20 °C/min in Ar atmosphere.

**Table 3.** DSC results of the AMZ4 powder, AB and RHT samples using a heating rate of 20 °C/min, where $T_g$, $T_x$, $\Delta T$, $\Delta H_x$, and $\Delta H_r$ are the onset temperature of glass transition, the onset temperature of crystallization, supercooled liquid region ($T_x$–$T_g$), enthalpy of crystallization, and enthalpy of relaxation, respectively. The amorphous fraction was calculated as the ratio $\Delta H^{AB}_x / \Delta H^{Powder}_x$.

| Sample | $T_g$ (°C) | $T_x$ (°C) | $\Delta T$ (°C) | $\Delta H_x$ (J/g) | $\Delta H_r$ (J/g) | Amorphous Fraction |
|--------|------------|------------|------------------|---------------------|---------------------|---------------------|
| Powder | 401 | 473 | 72 | 35.12 | N.A | 1 * |
| AB | 398 | 471 | 73 | 33.55 | 1.84 | 0.96 |
| RHT | 406 | 471 | 65 | 32.31 | 1.31 | 0.92 |

* We assume there is no crystal in the gas-atomized powder.

## 3.1. Structure and Fatigue Behavior of As-Built Alloy

The parameters used here were taken from reference [8], which led to a good combination of density and amorphous content, demonstrated by the resulting high compression and flexural strengths, and high wear resistance.

The XRD pattern of the AB sample (Figure 3a) shows an amorphous structure (within the detection limit of the XRD technique). However, back-scattered electron (BSE) images of the X–Z cross-section (Figure 3b,c) show the presence of tiny crystals in the HAZ. Crystal sizes were in the sub-micron range, which is similar to those reported in our previous studies [8,40,60]. The average value of hardness (HV1) from ten measurements was 451 ± 6. The amorphous fraction ($\Delta H^{AB}_x / \Delta H^{Powder}_x$) in the AB sample was calculated as 0.96. The oxygen content of the powder and the fabricated parts was measured in our previous study [8] as 1300 ppm and 1480 ppm, respectively (using the exact same batch of powder as in this study). Thus, almost 200 ppm of oxygen content was absorbed during the LPBF process.

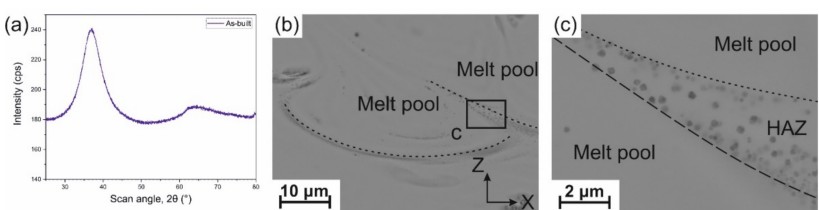

**Figure 3.** (**a**) XRD pattern of the AB sample from the X–Z cross-section; (**b**) back-scattered electron (BSE) image of the X–Z cross-section of the AB condition, and (**c**) higher magnification image of region c in (**b**).

Figure 4 shows the stress range versus the number of cycles to failure (S–N) curve of the specimens in the (AB) condition. The fatigue limit for the AB condition was measured as 175 MPa.

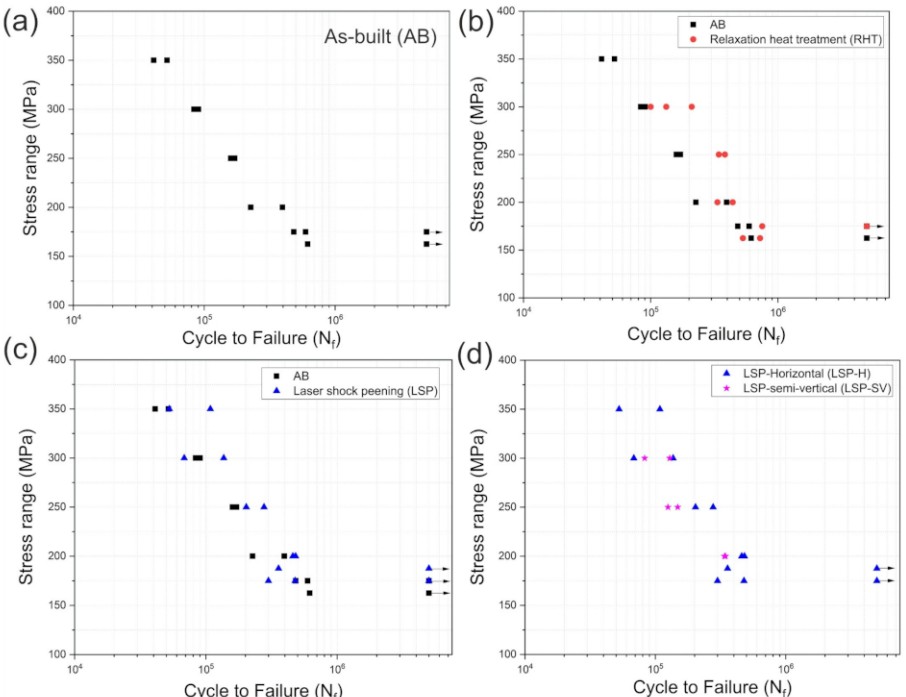

**Figure 4.** Stress range versus the number of cycles to failure (S–N) curve of (**a**) as-built (AB); (**b**) relaxation heat-treatment (RHT) vs. AB; (**c**) laser shock peening (LSP) vs. AB, and (**d**) LSP-horizontal (LSP-H) vs. LSP-semi-vertical (LSP-SV). The runout was set at $5 \times 10^6$ cycles, as shown by the arrows.

The fracture surfaces of two specimens tested at low and high loads, 162.5 MPa and 350 MPa, respectively, are presented in Figure 5. As can be seen in Figure 5b (low load), failure is caused by an open superficial LoF. However, the specimen tested at high load did not fail due to LOF defects, even though some of them were visible on the fracture surface. The crack nucleation could be due to shear band formation. In Figure 5c, due to the high loading level, crack propagation reached very quickly fast fracture stage III. The region associated with stage II crack growth, above the black dash-dotted curve in Figure 5d, was therefore restricted to a small area.

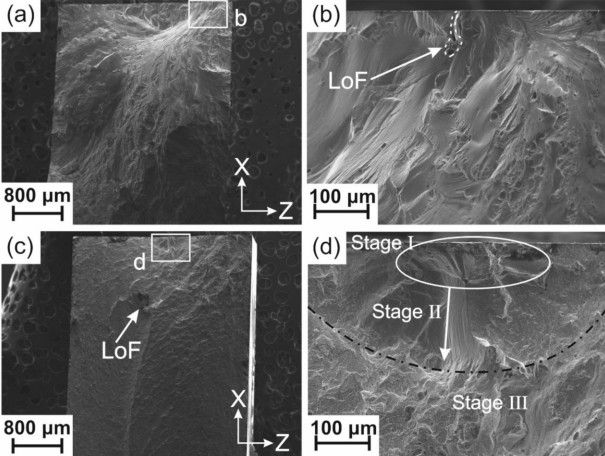

**Figure 5.** Fracture surfaces of two samples in the AB condition (**a,b**) at low load, 162.5 MPa, and (**c,d**) at high load, 350 MPa.

To further study the fracture surface in detail, higher magnification images taken from a sample tested at 200 MPa are shown in Figure S1b–d. Figure S1b,c represent a region

related to the fracture stage II. The fracture surface is not rough, and striations [7] are clearly visible. Each striation was formed during one or several loading cycle/s, and the crack propagation speed can be related to the size of the striation [7]. Figure S1d shows another zone, with dimple-like patterns [7], indicating a fast fracture region (stage III).

### 3.2. Effect of Relaxation Heat Treatment

The XRD pattern of a specimen after relaxation heat treatment for one hour at 320 °C (0.8 $T_g$) is presented in Figure 6a. In general, it shows an amorphous structure (within the detection limit of the XRD technique), but a tiny shoulder (shown by a black arrow) is also detected, which may come from the growth of pre-existing nanocrystals in the structure (see Figure 3b,c). The DSC curves of the AB and RHT samples are shown in Figure 6b. The hatched area in the inset of Figure 6b shows the relaxation enthalpy, $\Delta H_r$, which is higher for the AB sample. This means that the relaxation heat treatment was effective in the reduction of free volume. The characteristic thermal properties of the AB and RHT samples are given in Table 3. Figure 6c,d show the microstructure of the X–Z cross-section of the RHT sample. The presence of crystals in the HAZ is evident, and their size is larger than those in the AB condition. As a result of the higher crystalline fraction (8% instead of 4%, see Table 3), the hardness (HV1) was increased by 6% compared to the AB value, from $451 \pm 6$ to $478 \pm 10$.

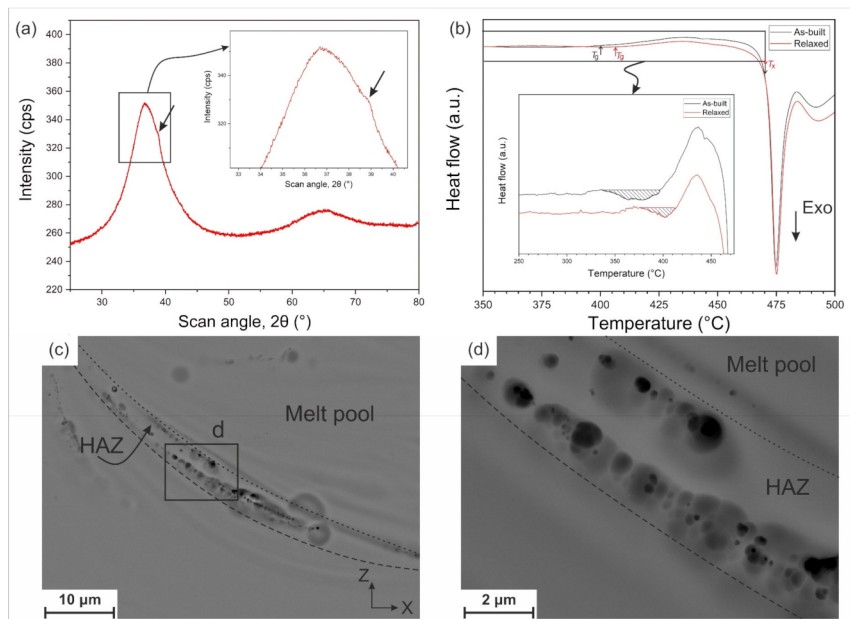

**Figure 6.** (**a**) XRD pattern of the RHT specimen from the X–Z cross-section showing a tiny shoulder in the inset; (**b**) DSC curves of the AB and RHT samples measured at a heating rate of 20 °C/min. The hatched area in the inset shows the enthalpy of relaxation for the two conditions, AB and RHT. (**c**) BSE image of the X–Z cross-section of the RHT condition, and (**d**) higher magnification image of region d in (**c**).

The fatigue results of the RHT specimens are presented in Figure 4b (red circles) and compared to the AB condition. At higher loads, the collected data for RHT specimens showed potential improved fatigue life, but due to the scatter of the results it could not be considered significant. However, for the stress range below 250 MPa, the results in both conditions were very similar. The fatigue limit of the RHT condition was consistently measured at the same level as that of the AB condition, namely 175 MPa.

Fracture surfaces of two specimens in the RHT condition tested at low and high loads, 162.5 MPa and 250 MPa, respectively, are shown in Figure 7. Similar to the AB condition at a low load, the crack started from a LoF exposed on the sample surface. No LoF was

detected in the vicinity of the initiation site of cracking for the sample tested at a higher load (see Figure 7c,d), which is again similar to the AB specimen tested at a high load (see Figure 5c,d).

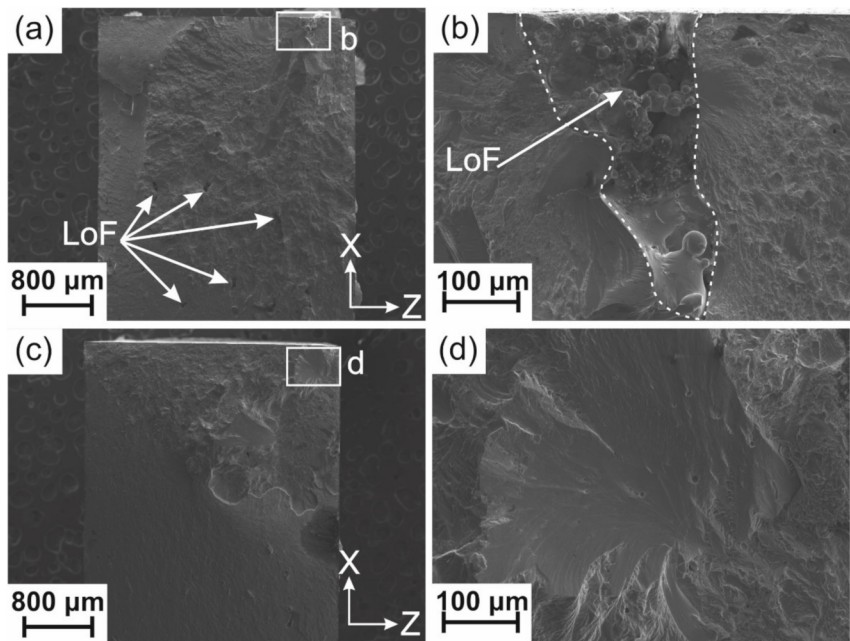

**Figure 7.** Fracture surface of two samples in the RHT condition (**a**,**b**) at low load, 162.5 MPa, and (**c**,**d**) at high load, 250 MPa.

### 3.3. Effect of LSP

The XRD pattern of an LSPed specimen (horizontal orientation) is presented in Figure S2. The amorphous pattern was similar to the AB sample and showed that the LSP treatment did not cause further crystallization (within the detection limit of the XRD technique). The surface hardness of the LSPed specimen was measured at $456 \pm 12$, i.e., similar to the AB state, considering the uncertainty of the measurement.

Figure 4c shows the fatigue results of LSPed specimens (blue triangles). LSP slightly increased the fatigue life of the specimens, especially at higher loads,) compared to the AB condition, and the fatigue limit was increased from 175 to 187.5 MPa. The 7% increase appeared as a slight potential improvement; considering the statistical scatter of the fatigue results, it was not considered to be significant.

Figure 8 shows the fracture surface of two LSPed samples at low and high loads, 175 MPa and 350 MPa, respectively. In both cases, the presence of a sub-surface LoF porosity explained crack initiation. These results were consistent with other samples tested at the same load.

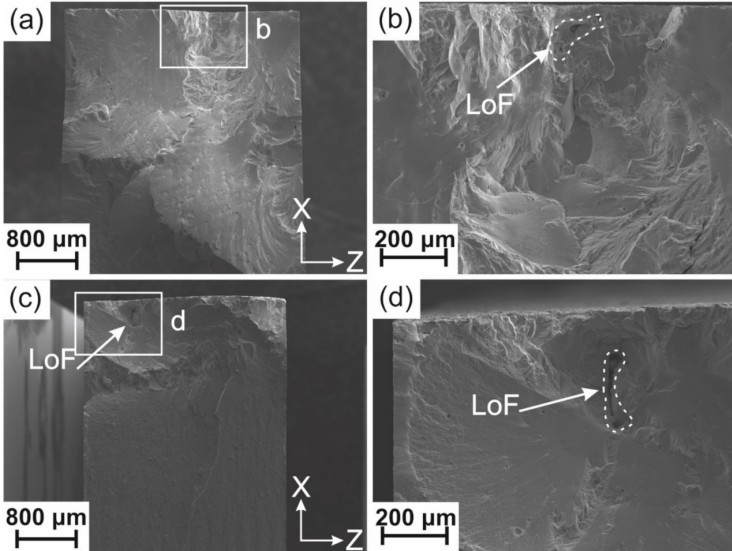

**Figure 8.** Fracture surface of two samples in the LSP-H condition (**a**,**b**) at low load, 175 MPa, and (**c**,**d**) at high load, 350 MPa.

### 3.4. Effect of Build Orientation

To investigate the effect of build orientation on the fatigue life of a Zr-based BMG fabricated via the LPBF process, six samples were produced in semi-vertical orientations. The small number of samples did not allow the full S–N curve the be covered and to determine the fatigue limit, but a fair comparison with the specimens fabricated in the horizontal orientation was possible. Figure 4d shows that the change in the build orientation (from horizontal to semi-vertical) did not bring any significant change in fatigue life at the stress levels of 250 and 200 MPa.

The fracture surface of the sample tested at a low load, 200 MPa (Figure 9a,b), showed no LoF close to the initiation site of the main crack, while at a higher load, 250 MPa, a large LoF did initiate the fatigue crack (see Figure 9c,d).

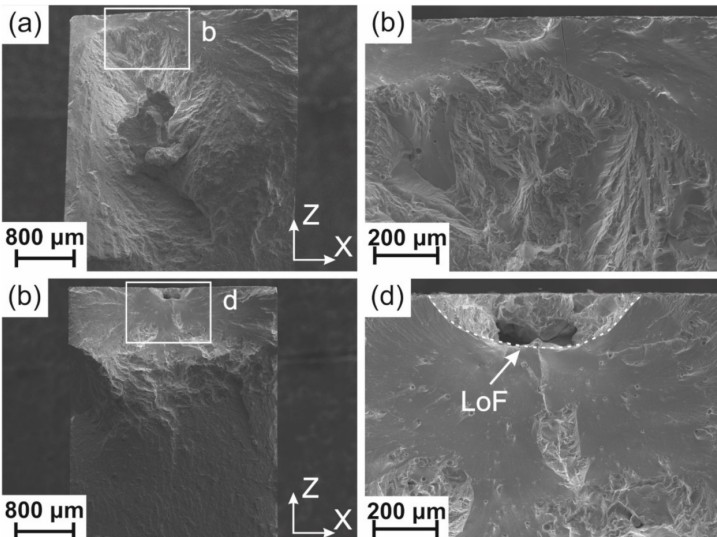

**Figure 9.** Fracture surface of two specimens in the LSP-SV condition (**a**,**b**) at low load, 200 MPa, and (**c**,**d**) at high load, 250 MPa.

## 4. Discussion

Although the XRD pattern of the AB sample (see Figure 3) indicates an amorphous structure, nanocrystals were formed in HAZs due to the local thermal cycles induced in these regions by the overlapping and adjacent track deposition. The conventional XRD method is not sensitive enough to detect low volume fractions of crystallization, as already noticed in references [8,12,40,60].

We recently studied crystallization of AMZ4 [67] and showed that the time to crystallization is in the order of 3 ms in the 750–800 °C temperature range, which effectively prevents the fabrication of fully amorphous AM parts. Bordeenithikasem et al. [9] showed that crystallization of AMZ4 results in an increase of hardness and brittleness, making the material prone to cracking, which is consistent with the results of reference [60].

As mentioned before, the measured fatigue limit in the AB condition is 175 MPa, which represents 20% of the UTS of AMZ4 reported in reference [40] fabricated with the same processing parameters as in the present study, i.e., a fatigue ratio of 0.2. The low value of UTS was also attributed to the presence of LoFs, which is inevitable in LPBF of AMZ4. Since the glass-forming ability of the commercial AMZ4 is low and it is very sensitive to crystallization, a low input energy should be used during the LPBF process to prevent extensive crystallization, which, inevitably, results in LoF formation. If one considers the UTS of the as-cast sample (1.7 GPa, reported in the datasheet of Heraeus [65]), the fatigue ratio of the LPBF sample would decrease to 0.11.

Using the X-ray micro-computed tomography technique (μCT), the overall porosity content of a sample (26 mm$^3$ in volume) fabricated with the same parameters as those used here, was quantified as 0.23%. Several LoFs identified by different colors were detected (see Figure S3). As shown in Figure 10a, the equivalent diameters of LoFs were mainly below 60 μm, but a few had a size exceeding 200 μm. Figure 10b shows the LoFs shape factors, which could be as high as 50. The combination of large LoFs with irregular shapes was very detrimental to the fatigue properties.

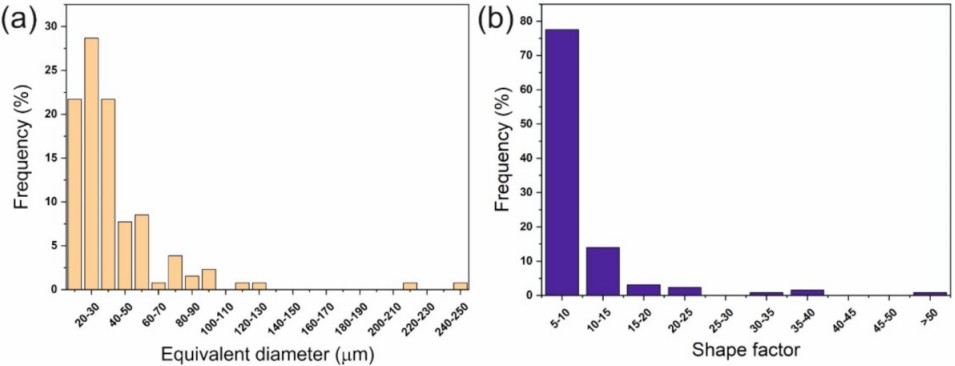

**Figure 10.** (**a**) Histogram of the equivalent diameter of LoFs, and (**b**) histogram of the shape factor of LoFs, where higher values represent more irregular and convoluted shapes.

Bending fatigue strengths of several Zr-based BMGs from the literature are presented in Table 4. Except for the AMZ4 investigated in the present study, other BMGs were fabricated via casting. Unfortunately, there is no data for the fatigue life of as-cast AMZ4, which could be used as a (defect free) reference for the present study. Table 4 indicates that the fatigue limit not only depends on the chemical composition, but also on the sample size, stress ratio (R), and type of bending test. In a review paper, Jia et al. [7] showed that surface quality, test conditions, environment, and cyclic frequency also impact the fatigue behavior of BMGs. It seems that bending fatigue tests (three-point or four-point) result in a lower fatigue limit. The absolute value of the fatigue limit of AMZ4 measured in this work is ranked among the lowest and is similar to the partially crystallized BMGs. This low value can be attributed to the combination of several reasons: (1) the

presence of porosity defects, such as LoFs, in the fabricated part; (2) the presence of a high level of oxygen content; (3) partial crystallization; and (4) the presence of TRS. The fatigue ratio is however comparable to stress-relieved $Zr_{44}Ti_{11}Ni_{10}Cu_{10}Be_{25}$ [26] and $Zr_{56.2}Cu_{6.9}Ni_{5.6}Ti_{13.8}Nb_{5.0}Be_{12.5}$ [68], which contained a ductile crystalline phase and was higher than Vitreloy 1 [22], for which the specimen size was close to the one used in the present study.

According to Yue et al. [22], at lower loads, fatigue cracks are initiated from defects occupying fairly large cross-section areas. The presence of near-surface LoF porosities on the fracture surface, especially at lower loads (Figure 5b, Figure 7b, and Figure 8), is evident. LoFs have irregular shapes and can locally cause stress concentration and act as initiation sites for fatigue cracks [41,43]. Therefore, LoFs largely contribute in reducing the fatigue limit of AMZ4 fabricated via LPBF and in increasing variability of fatigue life. The low UTS value of AMZ4 fabricated via LPBF was also related to the presence of LoFs in the near-surface (contour) region. Although the strategy of increasing the laser power in the near-surface region slightly increased the crystallized fraction, it was effective in reducing the LoF content, thereby significantly increasing the tensile strength (by 27%) [40]. Reducing porosity content by hot isostatic pressure (HIP) post-treatment, as done for crystalline alloys [69], is not an option for BMGs, due to the associated extensive crystallization, especially for AMZ4 exhibiting high critical heating and cooling rates [67].

In our previous study [8], the oxygen content of the part fabricated with the same powder batch and processing parameters as those used in the current study was measured as 1480 ppm. Best et al. [15,39] showed the detrimental effect of oxygen content on the fracture toughness in LPBF and as-cast samples and correlated this effect to the reduction of atomic mobility. The oxygen content is also known to accelerate the crystallization kinetics.

The presence of unwanted nanocrystals in the fabricated parts deteriorates the toughness of BMGs [70,71] by introducing brittle phase/s and/or intermetallics, which can act as stress risers and result in premature failure [60]. It has also been reported that crystallization drastically reduces the fatigue limit [18,72]. However, in our previous study [40], we showed that the effect of unwanted crystallization (up to 17 vol%) is less detrimental to tensile strength and impact toughness than the presence of defects, such as LoFs.

Sub-$T_g$ heat treatment (relaxation) resulted in the growth of nanocrystals found in the HAZs. This growth led to the formation of a tiny shoulder in the XRD pattern (see Figure 6). The value of $\Delta H_x$ for the RHT sample was lower than that of the AB condition, which confirms the increase in the crystalline fraction after relaxation. In addition, $\Delta T$ was reduced compared to the AB sample as a result of the increase in the crystalline fraction. Based on the hatched area in the inset of Figure 6b and the value of $\Delta H_r$ in Table 3, sub-$T_g$ heat treatment was effective in the reduction of the free volumes, i.e., relaxing the material. However, it was not enough to completely relax the material. Heat treatment for a longer time or at a higher temperature would however have increased the crystallization fraction, and dramatically reduced toughness.

Although relaxation treatments increased the crystalline fraction and despite its known detrimental effect on fatigue limit [72], no decrease was detected comparing RHT and AB conditions. As explained by Launey et al. [26], this could be attributed to the reduced amount of free volumes in RHT samples, inhibiting crack initiation. A 41% improvement in fatigue limit was then reported after relaxation (compared to the stress-relieved state), which should be connected to the absence of crystallization in these relaxation treatments.

The fatigue life of LSPed specimens was marginally improved at higher loads (see Figure 4c). Unchanged hardness values contrast with the expected softening typically induced by CRS [52,55]. This indicates that the stress state on the LSPed surface is still tensile, or slightly compressive. We can assume that this effect leads to the slight improvement observed in the fatigue life at higher loads. Measuring the residual stress state of BMGs with conventional XRD is complicated by the absence of sharp diffraction peaks. Furthermore, surface areas are not large enough for placing the strain gauges needed for assessing stresses with the hole-drilling method [48]. Numerical simulations [5,53] and/or

digital image correlation [53] would be required to measure the stress state, but this goes beyond the scope of our study. In crystalline alloys, such as SS316L [48], LSP results in plastic deformation of the near-surface region, leading to a change in residual stresses, but also in increased hardness and shrinking of porosities. Fatigue life improvement is then a consequence of all these effects. Comparatively, AMZ4 is not expected to undergo significant plastic deformation, which then leaves the porosity sizes and shapes quasi unchanged after LSP.

LSP-H specimens showed slightly higher fatigue life compared to that of the LSP-SV (see Figure 4d), which is attributed to the orientation of LoFs with respect to the applied load. LoFs with different orientations were shown in the µCT image, as illustrated in Figure S3 in the supplementary material. Figure 11a,b show schematics of two different orientations of a LoF defect, according to the chosen laser scan direction. In this study, the island scanning strategy may introduce LoFs elongated along either X or Y (Figure 11). In Figure 11c, the projected area of the LoF on a plane perpendicular to the loading direction is the same for both build orientations. However, Figure 11d shows that the projected area of the LoF for LSP-H is smaller. This indicates a smaller initial flaw size, which results in a slightly longer fatigue life. This reasoning is consistent with the findings of references [43,46].

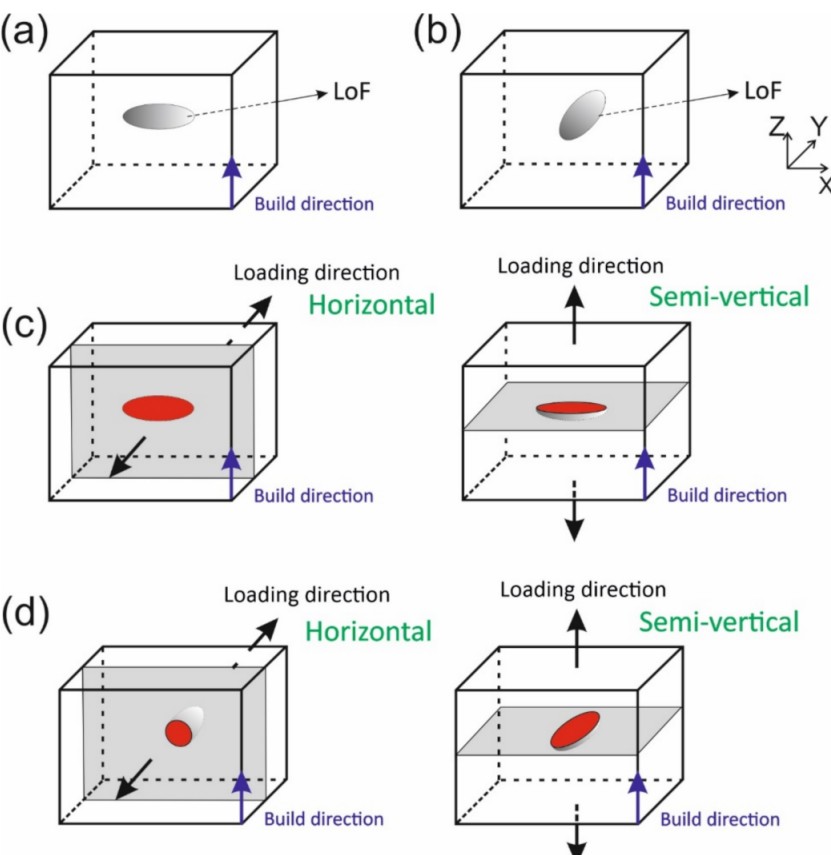

**Figure 11.** Illustrations of orientations of LoF in LPBF fabricated parts showing how build and loading directions may affect fatigue life: (**a**) LoF oriented in the X direction; (**b**) LoF oriented in the Y direction; (**c**) projected area of an LoF oriented in the X direction in a plane perpendicular to the loading direction; and (**d**) projected area of an LoF oriented in the Y direction in a plane perpendicular to the loading direction.

**Table 4.** Comparison of fatigue limits and ultimate tensile strength (UTS) for Zr-based BMGs, considering different bending tests and sample geometries.

| Material | UTS (MPa) | Geometry (mm) | Frequency (Hz) | R | Test Method | Fatigue Limit (MPa) | Fatigue Ratio | Runout Limit |
|---|---|---|---|---|---|---|---|---|
| $Zr_{59.3}Cu_{28.8}Al_{10.4}Nb_{1.5}$, AMZ4-AB | 880 [40] | $60 \times 10 \times 3$ | 30 | 0.1 | Three-point bending | 175 | 0.20 | 5 M |
| $Zr_{41.2}Cu_{12.5}Ni_{10}Ti1_{3.8}Be_{22.5}$, Vitreloy 1 [22] | 1920 | $55 \times 15 \times 5$ | 25 | 0.1 | Three-point bending | 225 | 0.12 | 10 M |
| $Zr_{41.2}Cu_{12.5}Ni_{10}Ti_{13.8}Be_{22.5}$, Vitreloy 1 [31] | 1900 | $50 \times 3 \times 3$ | 25 | 0.1 | Four-point bending | 152 | 0.08 | 20 M |
| $Zr_{50}Cu_{37}Al_{10}Pd_3$ [73] | 1899 | $25 \times 3.5 \times 3.5$ | 10 | 0.1 | Three-point bending | 631 | 0.33 | 10 M |
| $Zr_{56.2}Cu_{6.9}Ni_{5.6}Ti_{13.8}Nb_{5.0}Be_{12.5}$, composite * [68] | 1480 | $30 \times 3 \times 3$ | 25 | 0.1 | Four-point bending | 296 | 0.20 | 20 M |
| $Zr_{50}Cu_{40}Al_{10}$ ** [18] | 1821 | $26 \times 3 \times 3$ | 10 | 0.1 | Four-point bending | 144 | 0.08 | 10 M |
| $Zr_{50}Cu_{30}Al_{10}Ni_{10}$ ** [18] | 1900 | $26 \times 3 \times 3$ | 10 | 0.1 | Four-point bending | 194 | 0.10 | 10 M |
| $Zr_{50}Cu_{37}Al_{10}Pd_3$ ** [18] | 1899 | $26 \times 3 \times 3$ | 10 | 0.1 | Four-point bending | 109 | 0.06 | 10 M |
| $Zr_{44}Ti_{11}Ni_{10}Cu_{10}Be_{25}$, annealed [26] | 1900 | $85 \times 2 \times 2.3$ | 20 | 0.3 | Four-point bending | 550 | 0.29 | 20 M |
| $Zr_{44}Ti_{11}Ni_{10}Cu_{10}Be_{25}$, stress-relieved [26] | 1900 | $85 \times 2 \times 2.3$ | 20 | 0.3 | Four-point bending | 390 | 0.21 | 20 M |
| $Zr_{52.5}Cu_{17.9}Ni_{14.6}Al_{10}Ti_5$, Vitreloy 105 [74] | 1700 | $30 \times 3.5 \times 3.5$ | 10 | 0.1 | Four-point bending | 425 | 0.25 | 10 M |
| $Zr_{52.5}Cu_{17.9}Ni_{14.6}Al_{10}Ti_5$, Vitreloy 105 [75] | 1700 | $25 \times 2 \times 2$ | 25 | 0.1 | Four-point bending | 408 | 0.25 | 20 M |

* Ductile crystalline phase was introduced; ** partially crystallized.

The proposed post-processing strategies, RHT, LSP, and change of build orientation did not significantly improve the fatigue behavior of AMZ4. We therefore conclude that the first order defects to consider for fatigue life are those initially present in the powder or introduced during LPBF, and which cannot be removed by post-processing. Among those previously listed, it would mean the oxygen content, and the defect content, especially LoFs.

In future work, the amount of porosity could be reduced further by increasing the laser power in the near-surface and/or core regions. Although it will induce more crystallization, a compromise should be found between the reduction of LoF defects and the increase in the fraction of crystallization. Fatigue results could be compared with those obtained on reference as-cast samples, such as to determine the optimum processing conditions, which are not necessarily the same as those previously identified [8] when measuring static mechanical properties. AMZ4 powder feedstock with lower oxygen content, or an alternative Zr-based metallic glass powder with higher glass-forming ability and slower crystallization kinetics, could also be studied, with then the possibility to eliminate LoF defects without detrimental consequences on crystallization.

## 5. Conclusions

In this study, we investigated the fatigue behavior of a Zr-based bulk metallic glass (BMG), AMZ4, fabricated via laser powder-bed fusion (LPBF). While previous work reported on the fatigue behavior of BMGs prepared by other methods, it is to the authors' knowledge the first time fatigue testing is performed on a BMG fabricated by LPBF. Three strategies were implemented to improve the fatigue behavior of the as-built specimens: (1) relaxation heat treatment; (2) laser shock peening (LSP) treatment; and (3) change of build orientation. The following results can be outlined:

1.  The fatigue limit of AMZ4 fabricated via LPBF is lower than typical values reported for fully amorphous as-cast Zr-based BMGs, but comparable to partially crystallized ones;
2.  The reasons for the lower fatigue limit are correlated to the presence of defects, such as lack of fusion (LoF) defects, crystallization in the heat-affected zone (HAZ), high oxygen content, and the presence of tensile residual stresses (TRS). Among them, the presence of LoFs and the high oxygen content are concluded to have the largest impact. The exact effect of the powder oxygen content remains to be quantified;
3.  Although relaxation heat treatment (RHT) increased the crystalline fraction, it also slightly increased the fatigue life at higher loads ($\geq$250 MPa). This is attributed to the reduction of free volume and shear band formation, delaying crack initiation;
4.  LSP marginally increased the fatigue limit by 7%. Considering the scatter of fatigue results, the improvement was not considered to be significant. However, LSP improved the fatigue life at higher loads ($\geq$250 MPa) by a factor of two;
5.  Changing the samples' build orientations from horizontal to semi-vertical did not affect the fatigue behavior at lower loads but slightly decreased the fatigue life at higher loads (exceeding 250 MPa). The latter effect can be explained from LoFs orientation effects, with respect to the applied load.

Since the post-processing strategies used in this study did not significantly improve fatigue performance, optimal fatigue properties will result from an accurate compromise between the decrease in LoF porosity and the associated increase in the crystallized fraction. Future work will therefore focus on the gradual increase of laser power in the core and/or near-surface (contour) regions of the part. Another option will consist of testing powders with lower oxygen content.

**Supplementary Materials:** The following are available online at https://www.mdpi.com/article/10.3390/met11071064/s1, Figure S1: (a) Macroscopic image of the fracture surface of a sample tested at 200 MPa; (b) higher magnification image of region b in (a); (c) higher magnification image of (b) showing striation patterns; and (d) higher magnification image of region d in (a) shows dimple-like patterns as a result of rapid fracture, Figure S2: XRD pattern of laser shock peened (LSPed) sample

from the X–Z cross-section, Figure S3: 3D constructed image of an LPBF sample (as-built without any post-processing), where the Z-axis is the build direction of a sample fabricated horizontally and the force was applied in the X direction. Black arrows highlight LoFs oriented in the X-direction (or horizontal orientation), and red arrows highlight LoFs oriented in the Y-direction (or vertical orientation).

**Author Contributions:** Conceptualization, N.S. and R.E.L.; Data curation, A.P.; Formal analysis, N.S., M.H.-N., B.R., and A.P.; Investigation, N.S., M.H.-N., B.R., and A.P.; Methodology, N.S., M.H.-N., and A.P.; Project administration, R.E.L.; Resources, M.V. and R.E.L.; Supervision, R.E.L.; Validation, J.J. and R.E.L.; Visualization, N.S.; Writing—original draft, N.S.; Writing—review and editing, M.H.-N., B.R., J.J., A.P., M.V., and R.E.L. All authors have read and agreed to the published version of the manuscript.

**Funding:** This work was supported by the "PREcision Additive Manufacturing of Precious metals Alloys (PREAMPA)" project. The PREAMPA project is funded by the Swiss ETH domain, within the Strategic Focus Area on Advanced Manufacturing.

**Institutional Review Board Statement:** Not applicable.

**Informed Consent Statement:** Not applicable.

**Data Availability Statement:** The raw/processed data required to reproduce these findings cannot be shared at this time as the data also forms part of an ongoing study.

**Acknowledgments:** The generous support of the PX Group to the LMTM laboratory is highly acknowledged.

**Conflicts of Interest:** The authors declare no conflict of interest.

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
