# Peer review of "Fatigue Performance of an Additively Manufactured Zr-Based Bulk Metallic Glass and the Effect of Post-Processing"

_metals, doi:10.3390/met11071064_

Round 1

Reviewer 1 Report

It is known that the small size of the parts is mainly limited to the application of bulk metallic glasses. Additive manufacturing technologies may solve this problem. The authors of the paper “Fatigue performance of an additively manufactured Zr-based bulk metallic glass and the effect of post-processing” have investigated the influence of relaxation heat treatment, laser shock pinning, and changing of building orientation on the fatigue properties of Zr-based BMG. A large number of modern techniques were applied for the investigation of microstructure and properties. The paper is well written and may be accepted for publication. However, some additional details are required to be added accordingly following comments:

  1. The strength of the investigated samples is significantly lower than obtained in additive manufactured Zr-based BMG by other scholars (please, see 10.1016/j.addma.2018.03.023, 10.1016/j.jmst.2020.06.007, 10.1016/j.matdes.2020.108532. Why the authors did not make the optimization of the laser powder-bed fusion technology and investigated very defective samples?
  2. Minor suggestions are:
  • The temperatures Tg and Tx should be shown in DCS-curves by arrows;
  • The runout limit for determination of fatigue limit should be also added to Table 4.

Author Response

It is known that the small size of the parts is mainly limited to the application of bulk metallic glasses. Additive manufacturing technologies may solve this problem. The authors of the paper “Fatigue performance of an additively manufactured Zr-based bulk metallic glass and the effect of post-processing” have investigated the influence of relaxation heat treatment, laser shock pinning, and changing of building orientation on the fatigue properties of Zr-based BMG. A large number of modern techniques were applied for the investigation of microstructure and properties. The paper is well written and may be accepted for publication. However, some additional details are required to be added accordingly following comments:

The strength of the investigated samples is significantly lower than obtained in additive manufactured Zr-based BMG by other scholars (please, see 10.1016/j.addma.2018.03.023, 10.1016/j.jmst.2020.06.007, 10.1016/j.matdes.2020.108532. Why the authors did not make the optimization of the laser powder-bed fusion technology and investigated very defective samples?

Two of the studies mentioned by the reviewer are related to other Zr-based BMGs, not AMZ4 and none of the three investigated tensile properties of the fabricated parts. Although the first reference mentioned by the reviewer is about AMZ4 (10.1016/j.addma.2018.03.023), in our previous study (https://doi.org/10.1016/j.matdes.2020.109400), we could achieve a higher flexural strength by the same parameters used in the present study. Previously (https://doi.org/10.1016/j.addma.2020.101124), It has been shown that increasing the input energy resulted in better density, but increased the crystalline fraction and crystallization is known to be detrimental for the mechanical properties. However, in the last paragraph of the discussion, we mentioned “In future work, the amount of porosity could be reduced further by increasing the laser power in the near-surface and/or core regions. Although it will induce more crystallization, a compromise should be found between the reduction of LoF defects and the increase in the fraction of crystallization. Fatigue results could be compared with those obtained on reference as-cast samples, such as to determine the optimum processing conditions,  which are not necessarily the same as those previously identified [8] when measuring static mechanical properties. AMZ4 powder feedstock with lower oxygen content, or an alternative Zr-based metallic glass powder with the higher glass-forming ability and slower crystallization kinetics, could also be studied, with then the possibility to eliminate LoF defects without detrimental consequences on crystallization.” The following sentence is added to the manuscript, page 7.

“The parameters used here were taken from Ref [8] that led to a good combination of density and amorphous content, demonstrated by the resulting high compression and flexural strengths, and high wear resistance.”

Minor suggestions are:

The temperatures Tg and Tx should be shown in DCS-curves by arrows;

Thank you for noticing. They have been added in the revised manuscript

The runout limit for determination of fatigue limit should be also added to Table 4.

Thank you for your suggestion. It has been added in Table 4 of the revised manuscript.

Reviewer 2 Report

The paper presents fatigue studies of LPBF of BMGs. I have the following comments: There are a wide number of groups working on LPBF of BMGs. Citations [1-10] includes very restricted number of research groups where 4/10 are self citations. Please provide more relevant citations. Provide better labelling in Figure 4 (probably the most important figure in this work) so that readers don't have to read figure captions to know what (a)-(d) is representing. Also x-axes should be "Cycles to Failure" For sub-Tg annealing, the more indicative measure is the enthalpic recovery at the glass transition event (the overshoot "peak" at Tg). A larger peak means the sample is more structurally relaxed and the results are very obvious. The relaxation enthalpy reported here is a poor choice since the undershoot is often very difficult to see and baseline selection plays a huge role. With a noisy baseline as reported, the errors could compound. I understand the difficulty in performing fatigue tests but having only 2 samples per load condition, it is hard to make statistical correlations especially for brittle failure modes. Sub-Tg annealing should make the sample more homogeneous and more brittle (this is well known). The results show a bigger scatter in the RHT samples and hence proves that the annealing has caused embrittlement. The authors cannot make claims that RHT make the fatigue life better since it is within errors of the experiment. Likewise claims of increases in fatigue limit by LSP could not be made. The measurements the authors provide have merit and should be published but it should be in the form of "the data is presented" or "the collected data suggests xyz" where no generalizations beyond data scatter/grouping should be made.

Author Response

The paper presents fatigue studies of LPBF of BMGs. I have the following comments: There are a wide number of groups working on LPBF of BMGs. Citations [1-10] includes very restricted number of research groups where 4/10 are self citations. Please provide more relevant citations. Provide better labelling in Figure 4 (probably the most important figure in this work) so that readers don't have to read figure captions to know what (a)-(d) is representing. Also x-axes should be "Cycles to Failure". For sub-Tg annealing, the more indicative measure is the enthalpic recovery at the glass transition event (the overshoot "peak" at Tg). A larger peak means the sample is more structurally relaxed and the results are very obvious. The relaxation enthalpy reported here is a poor choice since the undershoot is often very difficult to see and baseline selection plays a huge role. With a noisy baseline as reported, the errors could compound. I understand the difficulty in performing fatigue tests but having only 2 samples per load condition, it is hard to make statistical correlations especially for brittle failure modes. Sub-Tg annealing should make the sample more homogeneous and more brittle (this is well known). The results show a bigger scatter in the RHT samples and hence proves that the annealing has caused embrittlement. The authors cannot make claims that RHT make the fatigue life better since it is within errors of the experiment. Likewise claims of increases in fatigue limit by LSP could not be made. The measurements the authors provide have merit and should be published but it should be in the form of "the data is presented" or "the collected data suggests xyz" where no generalizations beyond data scatter/grouping should be made.

Based on reviewer’s suggestion, the citations were reorganized in the revised version.

Fig. 4 was redesigned in the revised version.

We agree with the reviewer that Sub-Tg annealing makes the material more brittle, but as explained in the introduction “Launey et al. [26] performed different heat treatments on an as-cast Zr-based BMG below the glass transition temperature, Tg, to relieve the residual stresses and reduce the free volume in the material. They showed that structural relaxation resulted in improved fatigue strength, attributed to the higher resistance in crack initiation with lower free volume, in agreement with earlier work by Launey et al. [27], and with the associated increase in hardness [28].” That is why improvement in the fatigue limit is expected after Sub-Tg annealing.

The hardness results showed an increase in the hardness of the samples after Sub-Tg annealing, which is an indication of relaxation and a decrease in the free volume content. This result is in agreement with the reduction in the enthalpy of relaxation in the DSC experiment. We agree with the reviewer that the noise in the DSC measurement could have influenced the intensity, that is why we compared the two conditions qualitatively, not quantitatively. We used the same concept as https://doi.org/10.1016/j.msea.2019.138535 (please see figure below) for comparing the enthalpy of relaxation.

Concerning the fatigue results of RHT condition, based on the reviewer’s suggestion, it has been modified in the revised version as follow (line 279 in the revised version): “At higher loads, the collected data for RHT specimens show potential improved fatigue life, but due to the scatter of the results it cannot be considered significant.” In addition, the following sentence in the discussion has been removed in the revised version (line 413) “Up to two times improvement in fatigue life was measured at higher loads ( ≥250 MPa).”

Regarding the fatigue results of LSP condition, it was already mentioned in the text that “The 7% increase appears as a slight potential improvement, considering the statistical scatter of the fatigue results, it is not considered to be significant.”

Reviewer 3 Report

Comments:

  • The atmosphere of DSC should be cited in the caption of Fig. 2.
  • -Discussion of the reasons for the low fatigue limit of BMG due to high oxygen content and residual tensile stress is not supported by the results. I believe that the residual tension should be measured, but I understand the justification presented by the authors, and the effect of oxygen should be evaluated, changing its concentration.

Author Response

The atmosphere of DSC should be cited in the caption of Fig. 2.

Thank you for noticing. It has been added in the caption of Fig 2 and in the Materials and Method section of the revised version.

-Discussion of the reasons for the low fatigue limit of BMG due to high oxygen content and residual tensile stress is not supported by the results. I believe that the residual tension should be measured, but I understand the justification presented by the authors, and the effect of oxygen should be evaluated, changing its concentration.

In addition to the value of the oxygen content of the fabricated part, which was given in the text, the oxygen content of the powder was also added in the revised manuscript, which was reported in our previous study, and we used the same batch of powder and the same processing parameters for the fabrication of the fatigue samples in the present study (line 227). We agree with the reviewer that the oxygen measurements were not the result of this study, but the detrimental effect of oxygen on the crystallization and mechanical properties of BMGs have been proven (one paragraph in Introduction was dedicated to the effect of oxygen, line 104 to line 112). Therefore, we do not need to reconfirm its effect by performing more experiments.

Concerning the residual stress measurement, since the cross-section of the samples was small (3.5×60 mm2), we could not put a strain gauge (an area of 8×12 mm2 is required) and measure the residual stresses of the fatigue samples using the hole drilling method. It was explained in the text (line 422) as well “Measuring the residual stress state of BMGs with conventional XRD is complicated by the absence of sharp diffraction peaks. Besides, surface areas are not large enough for placing strain gauges needed for assessing stresses with the hole-drilling method ”. We have the hole drilling results of other samples with a different geometry, but it has been shown in the literature that the geometry and size have an influence on the level of residual stresses. That is why those results cannot be applicable in this case.

Round 2

Reviewer 1 Report

The authors have answered previous comments. The manuscript may be accepted for publication.

Author Response

Thank you very much.

Reviewer 2 Report

The authors have adequately addressed the concerns put forward by the reviewers and after some minor edits, should be acceptable for publications.

There are some formatting errors with respect to the citations.

Author Response

Thank you very much for noticing.

Please check the latest version. The citation format has been changed.

If still, some issues have remained, we kindly ask the reviewer to be slightly more specific.